# A Prospective Real-World Multi-Center Study to Evaluate Progression-Free and Overall Survival of Radiotherapy with Cetuximab and Platinum-Based Chemotherapy with Cetuximab in Locally Recurrent Head and Neck Cancer

**DOI:** 10.3390/cancers13143413

**Published:** 2021-07-08

**Authors:** Markus Hecht, Dennis Hahn, Philipp Wolber, Matthias G. Hautmann, Dietmar Reichert, Steffi Weniger, Claus Belka, Tobias Bergmann, Thomas Göhler, Manfred Welslau, Christina Große-Thie, Orlando Guntinas-Lichius, Jens von der Grün, Panagiotis Balermpas, Katrin Orlowski, Diethelm Messinger, Karsten G. Stenzel, Rainer Fietkau

**Affiliations:** 1Department of Radiation Oncology, Universitätsklinikum Erlangen, Friedrich-Alexander-Universität Erlangen-Nürnberg, 91054 Erlangen, Germany; Rainer.Fietkau@uk-erlangen.de; 2Comprehensive Cancer Center Erlangen-EMN (CCC ER-EMN), 91054 Erlangen, Germany; 3Deutsches Zentrum Immuntherapie (DZI), Universitätsklinikum Erlangen, 91054 Erlangen, Germany; 4Klinikum Stuttgart, Klinik für Onkologie, 70173 Stuttgart, Germany; d.hahn@klinikum-stuttgart.de; 5Klinik für Hals-, Nasen-und Ohrenheilkunde, Universitätsklinikum Köln, 50935 Köln, Germany; philipp.wolber@uk-koeln.de; 6Klinik und Poliklinik für Strahlentherapie, Universitätsklinikum Regensburg, 93053 Regensburg, Germany; matthias.hautmann@ukr.de; 7Medizinische Studiengesellschaft NORD-WEST GmbH, 26655 Westerstede, Germany; info@onkologie-westerstede.de; 8Gemeinschaftspraxis Dres. Weniger/Bittrich, 99085 Erfurt, Germany; dr.steffi.weniger@onkologie-erfurt.de; 9Klinikum der Universität München (A.ö.R.), Klinik für Strahlentherapie und Radioonkologie, 81377 München, Germany; Claus.Belka@med.uni-muenchen.de; 10SRH Wald-Klinikum Gera GmbH, II. Medizinische Klinik, 07548 Gera, Germany; bergmann.studienzentrum@wkg.srh.de; 11Onkozentrum Dresden/Freiburg, 01127 Dresden, Germany; goehler@onkozentrum.de; 12Klinikum Aschaffenburg, Hämato-Onkologische Schwerpunktpraxis, 63739 Aschaffenburg, Germany; m.welslau@onkologie-ab.de; 13Zentrum Innere Medizin Klinik III-Hämatologie, Onkologie, Palliativmedizin, Universitätsmedizin Rostock, 18057 Rostock, Germany; christina.grosse-thie@med.uni-rostock.de; 14Klinik für Hals-, Nasen-und Ohrenheilkunde, Universitätsklinikum Jena, 07747 Jena, Germany; Orlando.Guntinas@med.uni-jena.de; 15Klinik für Strahlentherapie und Onkologie, Klinikum der J.-W. Goethe-Universität Frankfurt a.M., 60596 Frankfurt, Germany; Jens.VonderGruen@kgu.de (J.v.d.G.); Panagiotis.Balermpas@usz.ch (P.B.); 16Merck Serono GmbH, Medical Affairs Oncology, 94293 Darmstadt, Germany; katrin.orlowski@merckgroup.com (K.O.); karsten.stenzel@merckgroup.com (K.G.S.); 17Prometris GmbH, 68219 Mannheim, Germany; Diethelm.Messinger@prometris.com

**Keywords:** HNSCC, cetuximab, re-irradiation

## Abstract

**Simple Summary:**

Despite recent developments in immune checkpoint inhibitors, the treatment of locoregionally recurrent head and neck squamous cell cancer (HNSCC) remains challenging. Prospective data comparing re-irradiation with systemic treatment are not available. The SOCCER trial represents a prospective non-interventional multicenter trial that enrolled patients with locoregionally recurrent HNSCC treated with cetuximab in combination with re-radiotherapy or chemotherapy. A total of 192 patients were analyzed. Radiotherapy combined with cetuximab had superior progression-free and overall survival compared to chemotherapy with cetuximab. This highlights the high efficacy of local re-radiotherapy in combination with cetuximab in patients with locoregionally recurrent HNSCC.

**Abstract:**

Treatment options of locoregional recurrent head and neck squamous cell cancer (HNSCC) include both local strategies as surgery or re-radiotherapy and systemic therapy. In this prospective, multi-center, non-interventional study, patients were treated either with platinum-based chemotherapy and cetuximab (CT + Cet) or re-radiotherapy and cetuximab (RT + Cet). In the current analysis, progression-free survival (PFS) and overall survival (OS) were compared in patients with locoregional recurrence. Four hundred seventy patients were registered in 97 German centers. After exclusion of patients with distant metastases, a cohort of 192 patients was analyzed (129 CT + Cet, 63 RT + Cet). Radiotherapy was delivered as re-irradiation to 70% of the patients. The mean radiation dose was 51.8 Gy, whereas a radiation dose of ≥60 Gy was delivered in 33% of the patients. Chemotherapy mainly consisted of cisplatin/5-flurouracil (40%) or carboplatin/5-flurouracil (29%). The median PFS was 9.2 months in the RT + Cet group versus 5.1 months in the CT + Cet group (hazard ratio for disease progression or death, 0.40, 95% CI, 0.27–0.57, *p* < 0.0001). Median OS was 12.8 months in the RT + Cet group versus 7.9 months in the CT + Cet group (hazard ratio for death, 0.50, 95% CI, 0.33–0.75, *p* = 0.0008). In conclusion, radiotherapy combined with cetuximab improved survival compared to chemotherapy combined with cetuximab in locally recurrent HNSCC.

## 1. Introduction

Patients with locally advanced head and neck squamous cell cancer (HNSCC) are routinely treated with chemoradiotherapy either alone or adjuvant after surgery. After chemoradiotherapy, approximately 20% of patients develop local recurrences [1,2,3]. Furthermore, survivors have a more than ten-fold increased risk of developing a secondary primary tumor in the head and neck compared to the normal population [4]. Treatment options for these patients include local approaches such as salvage surgery, re-radiotherapy or re-radiotherapy combined with chemotherapy or cetuximab. The classical systemic treatment option was chemotherapy according to the EXTREME scheme (platinum/5-flurouracil/cetuximab) [5]. Recently, inhibitors of programmed cell death protein 1 (PD-1) either alone or in combination with platinum-based chemotherapy were also approved for first-line treatment [6]. In a subgroup analysis of the Keynote-040 trial patients with local tumor recurrences, PD-1 inhibitors were not more efficient than second-line single agent chemotherapy [7]. Thus, the question of local or systemic treatment remains highly relevant even after the approval of several immune checkpoint inhibitors. Among these treatment options, salvage surgery showed a survival benefit and should be recommended as treatment of choice [8,9]. However, salvage-surgery is often not possible due to the morbidity of the procedure and the functional consequences, especially for swallowing function and speech. No clear criteria exist for treatment selection of re-radiotherapy or systemic therapy, as prospective trials are missing.

The current analysis of the SOCCER study compares radiotherapy with cetuximab and platinum-based chemotherapy with cetuximab in locally recurrent HNSCC. The aim of this analysis is to detect differences in progression-free and overall survival between both treatment schemes.

## 2. Materials and Methods

### 2.1. Patients

Patients with locoregional recurrence and/or distant metastases from squamous cell cancer of the oral cavity, oropharynx, hypopharynx, and larynx were eligible for this prospective, multi-center, non-interventional study. The treatment in the study had to be first-line treatment in the recurrent situation. As the study should represent the real-life situation, the clinicians’ assessment of patients’ ability to be treated and enrolled in the study was not limited to baseline Eastern Cooperative Oncology Group (ECOG) performance status or blood parameters. Tumor stages were evaluated according to TNM, 7th edition.

### 2.2. Study Design and Treatments

The SOCCER study was a prospective, non-interventional, multi-center study. Patients received cetuximab either in combination with platinum-based chemotherapy (CT + Cet) or radiotherapy (RT + Cet). The treatment decision was made by the treating physician. Cetuximab application was in line with the European Medicines Agency (EMA) marketing authorization. Dosing of cetuximab began with an initial dose of 400 mg per square meter body surface area, followed by weekly doses of 250 mg per square meter body surface area. In the RT + Cet group, cetuximab treatment started one week prior to radiotherapy and was continued until the completion of radiotherapy. In the CT + Cet group, cetuximab was administered simultaneous to chemotherapy and continued as maintenance therapy until disease progression.

### 2.3. Endpoints and Assessments

The current analysis included only patients with loco-regional recurrence without distant metastases. The primary endpoint of the study was the evaluation of tumor symptom burden in responders and non-responders and was previously reported [10]. The patients’ tumor symptom burden was assessed using a questionnaire with visual analogue scales (VAS) for pain, breathing, swallowing, speech, smelling, taste, physical activity, and overall health state, which is summarized to the overall VAS score [10]. Secondary endpoints included overall survival (OS) and progression-free survival (PFS). Survival times were calculated beginning from the time point of first cetuximab infusion. RECIST criteria version 1.1 were recommended for tumor response assessment. There was no central RECIST evaluation. The best overall response was categorized during/after treatment as followed: complete response (CR), partial response (PR), stable disease (SD), progressive diseases (PD), and not assessable (NA). The overall response rate (ORR) in the two treatment groups was calculated as the proportion of patients with CR or PR as the best overall response.

### 2.4. Study Oversight

The study was registered at ClinicalTrials.gov (identifier: NCT00122460, https://www.clinicaltrials.gov/ct2/show/NCT00122460, accessed on 5 July 2021). The institutional review board at the Friedrich-Alexander-Universität Erlangen-Nürnberg (number: 84_12 B) approved the trial as leading board and, subsequently, all local ethic committees also approved it. Written informed consent was obtained from all patients before enrollment. The trial was designed by the academic authors in collaboration with the sponsor (Merck Serono GmbH, Darmstadt, Germany).

### 2.5. Statistical Analysis

Baseline characteristics between the two treatment groups (RT + Cet vs. CT + Cet) were compared using the Student’s *t*-test in the case of continuous variables and the chi-square test in case of categorical variables. Kaplan–Meier estimates were used for all time-to-event variables (PFS, OS and TTF) to estimates the survival rates and the median survival time after start of therapy. Appropriate 95% confidence intervals (CI) were determined for the survival probabilities at various time points and for the median survival time. The log-rank test was applied to compare the two treatment groups regarding the time-to-event variables, and Cox proportional hazard methods were used to estimate the corresponding hazard ratios with 95% confidence intervals. In addition, backward selection procedures were applied to identify patient characteristic significantly associated (*p* < 0.05) with OS and to estimate the treatment group effect adjusted for those factors. The factors considered in the selection procedure were gender, age, body weight, ECOG at baseline, Charlson comorbidity score, location of primary tumor, baseline overall VAS tumor symptom score, duration since initial diagnosis, previous radiotherapy, previous chemotherapy concomitant to radiotherapy, smoking status, and alcohol consumption. The ORR between the two treatment groups were compared using logistic regression methods. All statistical tests performed were of an exploratory nature.

## 3. Results

### 3.1. Patients

A total of 470 patients were registered in 97 German centers between October 2012 and June 2019. Eighty-four patients had to be excluded as they violated one or more eligibility criteria (Figure 1). The most frequent eligibility criterion that was violated was the use cetuximab without concurrent radiotherapy or concurrent platinum-based chemotherapy. Additional, 191 patients with distant metastases were excluded from this analysis focusing on patients with locoregional recurrence only. Three patients who received both radiotherapy and chemotherapy were also excluded. All analyses were performed in 192 patients, of whom 129 received chemotherapy and cetuximab and 63 radiotherapy and cetuximab. The clinical characteristics of these 192 patients are given in Table 1. Patients in the RT + Cet group were significantly older (*p* = 0.003) and tended to have a worse Charlson Comorbidity Index (*p* = 0.089). Previous treatment consisted of surgery in 67% of the patients in both groups. Previous radiotherapy was performed in 70% in the RT + Cet group and 92% in the CT + Cet group (*p* < 0.001).

### 3.2. Treatment

In the RT + Cet group, radiotherapy was delivered conventionally fractionated in most patients (92%) (Table 2). The mean dose per fraction was 1.9 Gy, up to a mean total dose of 51.8 Gy. A dose of 60 Gy or higher was achieved in 33% of the patients. Treatment was completed as planned in 86% of the patients. In the CT + Cet group, the most frequently used combination was cisplatin + 5-flurouracil (40%) followed by carboplatin + 5-flurouracil. The mean chemotherapy treatment duration was 12.8 weeks. The mean duration of cetuximab therapy was 8.6 weeks in the RT + Cet group and 18.9 weeks in the CT + Cet group (*p* < 0.001).

### 3.3. Efficacy

The mean observation time of the patients was 10.8 months in the RT + Cet group and 8.5 months in the CT + Cet group. One patient in the CT + Cet group was excluded due to missing follow up. Among the 192 included patients 156 PFS events and 130 OS events occurred.

PFS was significantly prolonged in the RT + Cet group (Figure 2A). The estimated 1-year PFS rate was 28% (95% confidence interval (CI), 16–41) in the RT + Cet group compared to 7% (95% CI, 3–13) in the CT + Cet group. Median PFS was 9.2 months (95% CI, 7.5–10.0) in the RT + Cet group and 5.1 months (95% CI, 4.3–6.0) in the CT + Cet group. The hazard ratio for disease progression or death was 0.40, (95% CI, 0.27–0.57, *p* < 0.001) for patients treated with RT + Cet versus CT + Cet.

OS was also significantly prolonged in the RT + Cet group (Figure 2B). One-year estimates for OS were 53% (95% CI, 38–66) in the RT + Cet group and 28% (95% CI, 19–36) in the CT + Cet group. Median OS was 12.8 months (95% CI, 9.2–18.5) in the RT + Cet group and 7.9 months (95% CI, 7.0–9.7) in the CT + Cet group. The hazard ratio for death was 0.50 (95% CI, 0.33–0.75, *p* = 0.0008) for patients treated with RT + Cet versus CT + Cet.

As these analyses of PFS and OS contained 29 patients without prior radiotherapy, subgroup analyses of patients with and without prior radiotherapy were performed. The PFS and OS benefit in the RT + Cet group remained statistically significant both in the groups with and without prior radiotherapy (Appendix A).

The treatment effect remained statistically significant (*p* = 0.024) in a backward selection procedure using multivariable Cox regression methods considering covariates which were also significantly associated with OS (Appendix A).

ORR in the RT + Cet group was 41% (95% CI, and significantly higher than the observed 26% in the CT + Cet group (*p* = 0.0282, Appendix A). CR, as the best overall response during therapy, was detected in 14% and 6% and PR in 27% and 19% in the RT + Cet and CT + Cet group, respectively.

In a subgroup analysis of the RT + Cet group, patients with a radiation dose of 60 Gy or higher were compared to patients with less than 60 Gy (Figure 3). The median PFS of 11.5 months in the subgroup ≥60 Gy was significantly longer than the PFS of 8.0 months in the subgroup <60 Gy (hazard ratio 0.49, 95% CI, 0.23–0.95, *p* = 0.0445). The median OS was 12.6 months in the subgroup <60 Gy compared to 15.9 months in the subgroup ≥60 Gy (hazard ratio 0.71, 95% CI, 0.32–1.48, *p* = 0.3664).

## 4. Discussion

Treatment options for patients with loco-regional recurrent HNSCC include salvage surgery, re-radiotherapy, and systemic therapy. Salvage surgery frequently is not possible due to its treatment-related morbidity. In this subgroup analysis of the SOCCER study, the classical first line chemotherapy consisting of platinum, 5-flurouracil, and cetuximab was compared to radiotherapy and cetuximab.

In the SOCCER study, both PFS and OS were significantly prolonged in patients treated RT + Cet compared to CT + Cet. This resulted in a clearly increased one-year OS rate of 53% in the RT + Cet group compared to 25% in the CT + Cet group.

Even though there was no randomized design, the SOCCER study is, to our knowledge, the only prospective comparison of radiotherapy with cetuximab and chemotherapy with cetuximab in patients with locoregional tumor recurrence. The non-randomized design leads to some imbalances in the patients’ characteristics. Patients in the RT + Cet group were older and tended to have more co-morbidities, which are confounding factors in disfavor of the RT + Cet group. The time interval since initial diagnosis was significantly longer in the RT + Cet cohort, whereas this had no impact on overall survival in the univariate analysis. Previous radiotherapy was performed in 70% in the RT + Cet group compared to 92% in the CT + Cet group. Consequently, this study is not a pure re-irradiation study. However, in separate subgroup analysis of patients with and without prior radiotherapy, a significant benefit regarding PFS and OS was detected in both groups favoring the combination RT + Cet. Treatment parameters of radiotherapy with a mean total dose of 51.8 Gy and only 33% of patients receiving total doses of ≥60 Gy reflect typical procedures of re-irradiation. A further limitation is that the p16 status was not available, which is an important prognostic factor in radiotherapy of HNSCC [11]. A further factor is that the combination of re-irradiation and cetuximab is considered as curative, and chemotherapy and cetuximab as palliative. This may be a bias in patient selection for both treatments that can hardly be detected. The major strength of the study is the prospective enrollment of a high patient number of patients (192) with locoregional recurrence. Another strength is that it reflects treatment in non-selected patients, as there were no limitations to ECOG performance score or blood parameters. Furthermore, the enrollment in a large study group containing 55 centers reflects an unbiased view of nationwide treatment.

The one-year OS rate of 53% in the radiotherapy and cetuximab group is comparable to previous studies combining cetuximab with re-radiotherapy [12,13]. The ORR of the radiotherapy and cetuximab group of 41% was slightly lower compared to these previous studies. Instead of cetuximab, cisplatin is also used frequently concomitant to re-radiotherapy. The efficacy of concomitant cisplatin application is probably similar to concomitant cetuximab [14]. However, as many of these patients had pretreatment with cisplatin, a chemotherapy-free treatment with concomitant cetuximab probably reduces treatment toxicity [15]. The mean radiation dose in this study was 51.8 Gy. In general, doses above 60 Gy are widely recommended for adequate efficacy of re-irradiation [9,16]. In a subgroup analysis of SOCCER patients treated with ≥60 Gy, the PFS significantly improved to 11.5 months (compared to 8.0 months if <60 Gy), without significant prolongation of OS. In a recent multicenter retrospective analysis, radiation doses of at least 66 Gy especially prolonged OS [16], which demands a further dose escalation. A further open question in the setting of re-irradiation is the comparison of concomitant treatment with cetuximab or cisplatin. Even though concomitant cisplatin was superior to cetuximab in first line treatment [17], the co-morbidities of patients requiring re-irradiation may lead to a different result in this setting.

The one-year OS rate of 25% of the chemotherapy and cetuximab group is lower than the approximately 40% in the phase III EXTREME trial [5]. The ORR of 26% is also lower than 36% in the EXTREME trial. This effect becomes even more evident when these results are compared to the more recent TPExtreme trial comparing docetaxel/cisplatin/cetuximab to cisplatin/5-flurouracil/cetuximab. The control arm of this trial reached a one-year OS of 56% and the study arm 59% [18]. This may be a consequence of the real-life population in the SOCCER study also including patients with worse ECOG performance scores and more comorbidities. Furthermore, the current analysis includes patients with locoregional tumor recurrence only, whereas both randomized trials also included patients with only distant metastases that typically have a better prognosis.

During the last few years, PD-1 inhibitors became the standard treatment in recurrent HNSCC. In the Keynote-048 trial, pembrolizumab alone (PD-L1 CPS ≥ 1 cohort) or in combination with chemotherapy was superior to cisplatin/5-flurouracil/cetuximab [19,20]. As the trial included both patients with recurrent and/or metastatic disease, pembrolizumab was approved for both types of disease as a first line treatment. However, in subgroup analyses of patients with locoregional recurrence, pembrolizumab monotherapy did not differ from cisplatin/5-flurouracil/cetuximab regarding OS even in selected patients (PD-L1 CPS ≥ 1) [19,20]. In these patients, the OS benefit of cisplatin/5-flurouracil/pembrolizumab compared to cisplatin/5-flurouracil/cetuximab was also lower compared to patients with metastatic disease. In a secondary analysis of the second line Keynote-040 study, pembrolizumab only prolonged survival compared to second line chemotherapy in patients with metastatic disease, but not in patients with local recurrence [7]. Furthermore, in this secondary analysis, only patients with previous radiotherapy mainly had a benefit from pembrolizumab. This may be a consequence of the immune modulating effects of radiotherapy [21], which also highlights this treatment sequence.

## 5. Conclusions

Taken together, in the SOCCER study patients receiving radiotherapy combined with cetuximab showed prolonged survival rates compared to patients receiving chemotherapy combined with cetuximab. This highlights the role of local treatment strategies in patients with sole locoregional tumor recurrences.

## Figures and Tables

**Figure 1 cancers-13-03413-f001:**
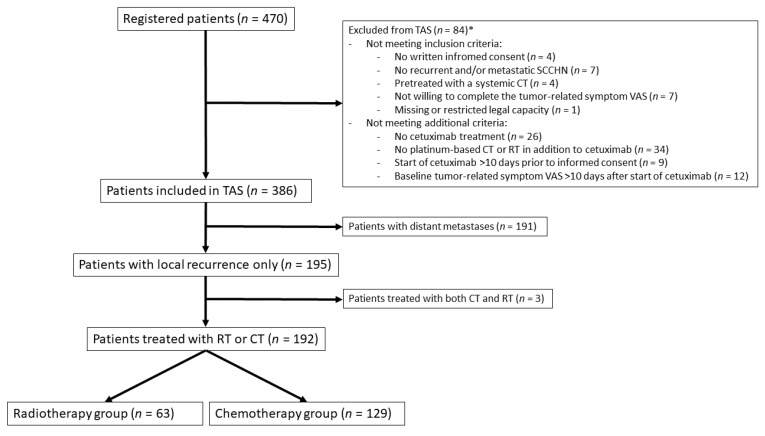
Consort diagram: CT, Chemotherapy; RT, Radiotherapy. (* Some patients violated more than one criterion).

**Figure 2 cancers-13-03413-f002:**
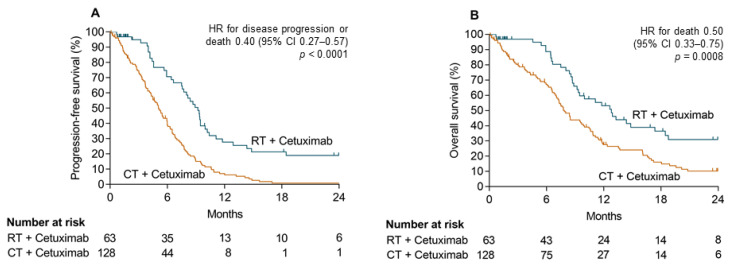
Survival analyses. Kaplan–Meier curves for progression-free survival (**A**) and overall survival (**B**) for all patients with locoregional recurrent HNSCC treated with radiotherapy and cetuximab (RT + Cetetuximab) or platinum-based chemotherapy and cetuximab (CT + Cetuximab). One patient treated with CT + Cetuximab was excluded due to missing follow up.

**Figure 3 cancers-13-03413-f003:**
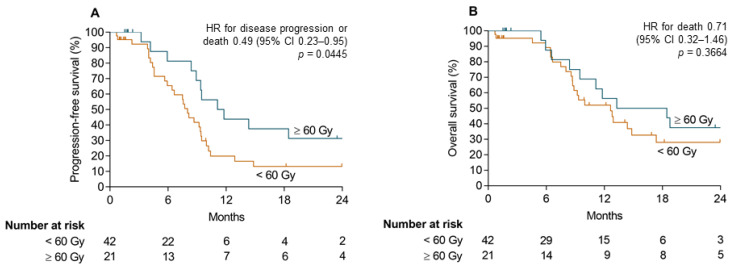
Survival analyses of RT + Cet group according to cumulative radiation dose. Kaplan–Meier curves for progression-free survival (**A**) and overall survival (**B**) for patients with cumulative radiation doses of at least or below 60 Gy.

**Table 1 cancers-13-03413-t001:** Patients’ characteristics.

Patients’ Characteristics	RT + Cet (*n* = 63)	CT + Cet (*n* = 129)	*p*-Value
**Age at Study Inclusion** [years], mean ± SD	66.6 ± 9.3	62.2 ± 9.5	0.003
**Weight** [kg], mean ± SD	66.2 ± 12.6	66.3 ± 13.0	0.959
**Sex**, *n* (%)			0.938
Female	13 (21)	26 (20)	
Male	50 (79)	103 (80)	
**Location of Primary Tumor** *, *n* (%)			
Oropharynx	20 (32)	34 (26)	0.436
Hypopharynx	14 (22)	27 (21)	0.838
Larynx	5 (8)	23 (18)	0.068
Oral cavity	21 (33)	43 (33)	1.000
Other location	6 (10)	12 (9)	1.000
**Stage of Disease at Initial Diagnosis**			0.396
I	4 (6)	12 (9)	
II	9 (14)	12 (9)	
III	11 (17)	15 (12)	
IV	37 (59)	87 (67)	
unknown	2 (3)	3 (2)	
**Prior Therapy**, *n* (%)			
Radiotherapy (with or without concomitant chemotherapy)	44 (70)	119 (92)	<0.001
Chemotherapy concomitant to radiotherapy	28 (44)	81 (63)	0.0134
Surgery	42 (67)	87 (67)	0.915
**Time Interval since Initial Diagnosis**, median (years)	2.2	1.2	<0.001
**Charlson Comorbidity Index** at study inclusion, *n* (%)			0.089
0	25 (40)	70 (54)	
1	12 (19)	25 (19)	
>1	26 (41)	34 (26)	
**ECOG Performance Status** at treatment initiation, *n* (%)			0.909
0	11 (18)	20 (17)	
1	38 (62)	74 (62)	
≥2	12 (20)	23 (19)	
**Alcohol Consumption**, *n* (%)			0.061
Never	18 (29)	26 (20)	
Several times per month	18 (29)	26 (20)	
Several times per week or daily	16 (25)	31 (24)	
Missing	11 (17)	46 (36)	
**Smoking Habits**, *n* (%)			0.247
Never smoked	20 (32)	35 (27)	
Former smoker	27 (43)	45 (35)	
Current smoker	16 (25)	48 (38)	
**Pack Years** of former/current smoker, mean ± SD	35.8 ± 24.7	37.5 ± 31.7	0.775

ECOG: Eastern Cooperative Oncology Group; SD: standard deviation; *: Multiple locations per patient possible.

**Table 2 cancers-13-03413-t002:** Treatment.

Treatment	RT + Cet (*n* = 63)	CT + Cet (*n* = 129)	*p*-Value
**Radiotherapy**			
Conventional fractionation, n (%)	58 (92)		
Dose per fraction [Gy], mean ± SD	1.9 ± 0.1		
Total dose			
<60 Gy, *n* (%)	42 (67)		
≥60 Gy, n (%)	21 (33)		
Mean ± SD [Gy]	51.8 ± 15.4		
Location			
Local relapse	44 (70)		
Lymph node	21 (33)		
Duration of radiotherapy [weeks], mean ± SD	6.6 ± 2.2		
Radiotherapy completed as planned, *n* (%)	54 (86)		
**Chemotherapy**			
Cisplatin + 5-Flurouracil, *n* (%)		52 (40)	
Carboplatin + 5-Flurouracil, *n* (%)		37 (29)	
Carboplatin, *n* (%)		18 (14)	
Cisplatin, *n* (%)		12 (9)	
Carboplatin + Paclitaxel, *n* (%)		7 (5)	
Other, *n* (%)		3 (2)	
Duration of chemotherapy [weeks], mean ± SD		12.8 ± 10.6	
**Cetuximab**			
Total duration of Cetuximab treatment [weeks], mean ± SD	8.6 ± 7.7	18.9 ± 19.6	<0.001
Cetuximab maintenance performed, *n* (%)	8 (13)	47 (36)	0.001

Gy, Gray; SD, standard deviation.

## Data Availability

Data on subjects’ level are made available on request after permission of the funding company.

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
