# Peer review of "A Prospective Real-World Multi-Center Study to Evaluate Progression-Free and Overall Survival of Radiotherapy with Cetuximab and Platinum-Based Chemotherapy with Cetuximab in Locally Recurrent Head and Neck Cancer"

_cancers, 2021, doi:10.3390/cancers13143413_

Round 1
Reviewer 1 Report
The authors present the data of the prospective non-interventional SOCCER study evaluating the role of re-radiation (RT) in combination with cetuximab vs. chemotherapy (CTX) in combination with cetuximab in recurrent head and neck cancer. In total, the study group analyzed 192 patients (129 with chemotherapy and cetuximab and 63 with radiation and cetuximab). Interstingly, patients with RT+Cetuximab had a significant prolonged PFS and OS compared to patients with CTX+Cetuximab. The paper is well written and the authors addressed a clinical relevant problem and unresovled question. The presented study is the largest study in this setting so far and the data is very impressive. However, there is one minor point the authors should address:
In the CTX+Cetuximab group 92% of patients had received radiation before the enrollment into the trial whereas only 70% of patients in the RT+Cetuximab group had received radiatio. This difference was statistically significant. The relative high number of non-irradiated patients in the RT+Cetuximab group could have a major impact on the results. The authors should include the radiation before the enrollment in the study in their multivariate anaylsis to exclude the impact of this imbalance. In addition, a subgroup analysis of the patients, which received prior radiation could be helpfull to clear that point.
I also would recommend to include a Forrest blot of the multivariate analysis into the manuscript. Unfortunately the supplemental table of the results was not available for the reviewer, only the supplemental table.
Reviewer 2 Report
This study aim to investigate the role of RT+Cet and CT+Cet (platinum based) combination in the setting of locally recurrent head and neck cancer. The comparison between radiotherapy with cetuximab and platinum-based chemotherapy with cetuximab was done.The results suggested bio-RT can achieve better survival. However, there are some critical issues in this study, making it less convincing.
1. The rationale of this study should be clearly described in the introduction.
2. What is the clinical stage distribution in these two groups?
3. The case number of patients in every primary site are quite small. In addition, the baseline characters of these group would be quite heterogenous.
4. No patient received prior CCRT?? There are 129 patients underwent prior operation, this is concerning that no risk factors (pathology characters, like positive margin....) for further CCRT.
5. The treatment decision was made by the physician. And the percentage of prior radiotherapy is higher in CT+Cet group. Further analysis to address the potential bias (selection bias) should be performed?
Reviewer 3 Report
Dear authors,
you submitted a non-interventional trial that compares Cetuximab+ (re-)irradiation with Cetuximab+ chemotherapy in patients with recurrent, locoregional head and neck cancer. The presented results of the superiority of Re-RT+ Cet against Cet+Chemo are not surprising as chemotherapy alone is regarded as palliative treatment, whereas reirradiation+Cet is considered curative. That is why I would heave preferred the question Re-RT+Cet vs. Re-RT+ cisplatin in patients with locoregional recurrent disease.
The decision between Re-RT and chemotherapy is most commonly done on the possibility of Re-RT, which is often impossible due to increased tissue toxicity because of prior irradiation (and of course patients comorbidities and preference) and not on the physicians choice as you stated. At least in our centre Re-RT is suggested whenever it is possible in comparison with chemotherapy because of its curative approach.
I would at least recommend to include this into the discussion sections and not present these two treatment strategies as equal approaches. This is also supported by the patients characteristics. In the Cet+Chemotherapy group 92% have been irradiated before, whereas in the Cet+RT group only 70% have been irradiated.
Can you further give information about how long ago the first RT was performed in the RT-Cet arm? As the time span between the two radiotherapy also influences the possibility of Re-RT.
Despite these issues I suggest acceptance of this study after major revisions because of the novelty of the comparison of these treatment arms and also because I find the results regarding the comparison of OS and PFS in patients receiving 60Gy and 52Gy interesting.
Good luck!
Round 2
Reviewer 2 Report
The quality of revised manuscript has significant improved. Some critical issues were answered and clarified properly in the revised manuscript.
Reviewer 3 Report
Dear authors,
as you have answered all my major revisions sufficiently I support publication of your study in Cancers.
Yours sincerely